# Characterization of Infants’ General Movements Using a Commercial RGB-Depth Sensor and a Deep Neural Network Tracking Processing Tool: An Exploratory Study

**DOI:** 10.3390/s22197426

**Published:** 2022-09-29

**Authors:** Diletta Balta, HsinHung Kuo, Jing Wang, Ilaria Giuseppina Porco, Olga Morozova, Manon Maitland Schladen, Andrea Cereatti, Peter Stanley Lum, Ugo Della Croce

**Affiliations:** 1Department of Electronics and Telecommunications, Politecnico di Torino, 10129 Torino, Italy; 2Department of Biomedical Engineering, The Catholic University of America, Washington, DC 20064, USA; 3Department of Biomedical Sciences, University of Sassari, 07100 Sassari, Italy; 4Children’s National Hospital, Washington, DC 20010, USA; 5Department of Rehabilitation Medicine, Georgetown University Medical Center, Washington, DC 20057, USA

**Keywords:** markerless, RGB-D, general movements, infant movement analysis, movement disorders

## Abstract

Cerebral palsy, the most common childhood neuromotor disorder, is often diagnosed through visual assessment of general movements (GM) in infancy. This skill requires extensive training and is thus difficult to implement on a large scale. Automated analysis of GM performed using low-cost instrumentation in the home may be used to estimate quantitative metrics predictive of movement disorders. This study explored if infants’ GM may be successfully evaluated in a familiar environment by processing the 3D trajectories of points of interest (PoI) obtained from recordings of a single commercial RGB-D sensor. The RGB videos were processed using an open-source markerless motion tracking method which allowed the estimation of the 2D trajectories of the selected PoI and a purposely developed method which allowed the reconstruction of their 3D trajectories making use of the data recorded with the depth sensor. Eight infants’ GM were recorded in the home at 3, 4, and 5 months of age. Eight GM metrics proposed in the literature in addition to a novel metric were estimated from the PoI trajectories at each timepoint. A pediatric neurologist and physiatrist provided an overall clinical evaluation from infants’ video. Subsequently, a comparison between metrics and clinical evaluation was performed. The results demonstrated that GM metrics may be meaningfully estimated and potentially used for early identification of movement disorders.

## 1. Introduction

Cerebral palsy (CP) is the clinical description given to a constellation of neuromotor impairments stemming from perinatal brain injuries such as periventricular leukomalacia, intracerebral hemorrhage, infection, and infant stroke [1]. A systematic review and meta-analysis [2] estimated the worldwide prevalence of CP at 2.11 births per 1000. Subsequent studies of various populations in Africa [3] Asia [4], and North America [5] suggest that the prevalence of CP is on the rise, at a rate of more than three per 1000, a phenomenon which may be due to the increasing likelihood of survival of early, preterm infants [6]. The average age for diagnosis of CP is 12–24 months in high-income countries and as late as five years in less well-resourced countries [7]. There are many reasons for diagnostic delay: the lack of definitive biomarkers for CP and definitive signs on traditional clinical examination, reluctance to communicate what might be a false positive to parents and triggering grief, uncertainty, and stigma, as well as the absence of curative treatments [8]. Arguably, the greatest boon to early identification of infants with CP has been the validation and dissemination of the general movement assessment, GMA. This instrument came into being as the understanding of the significance of infants’ spontaneous movements increased during the latter decades of the 20th century [9,10]. Two patterns in particular, predominantly cramped-synchronized movements and the absence of fidgety movements at three to five months of age reliably predict a later finding of CP [11]. Despite the power of the GMA in early identification, its key mechanism of gestalt pattern recognition (from video) requires a significant investment in assessor training and validation [12], making the GMA challenging to implement broadly across clinical practices. Early intervention depends on early identification, which suggests the need for a widely disseminated screening process, which is not found in the current approach to delivering CP care [8]. Families have been identified as the cornerstone of early intervention [13] and exploration of a more integral role for families in neurodevelopmental monitoring and therapy may result in earlier detection of developmental delays as well as earlier application of appropriate therapies. Engagement of families of infants with CP whose early signs of impairment are subtle may be particularly helpful given that these infants have been shown to be at greater risk for not receiving early diagnosis and intervention than are more profoundly affected infants [14]. Computationally assisted screening procedures likewise suggest a way to manage the increased clinical workflow that would result from broader application of neuromotor assessment among infants. Marker-based, multi-camera, 3D analysis of infant movement has been used to detect both upper [15] and lower extremity movements [16] correlated with GMA assessment of CP. Given that marker-based systems typically require multiple cameras and a laboratory setting; an accurate and reliable markerless computer vision approach that can be operationalized in either the home or clinic setting may make screening more widely accessible. Markerless computer vision technology further preserves the non-intrusive character of the GMA, leveraging, as does the GMA itself, video to assess an infant’s movements unhampered by markers or other sensing devices [12].

Computer vision techniques to automate the analysis of infant movements captured on 2D video have been under exploration for over a decade [17,18]. 3D recordings, however, may provide added value through higher spatial resolution, depth information, and higher accuracy and reliability; however, exploration has been limited by high technology cost and computational overhead [19]. Markerless computer vision systems have the ability to implement a kinematic model [20,21] and have been presented as a promising alternative to marker-based systems [22].

Avoiding markers may be particularly appropriate in the case of infants where they may be poorly tolerated and, as a result, introduce movement patterns that are not part of the infant’s typical GM repertoire [12].

Some markerless systems have employed a multi-camera approach [23] but a more accessible and practical solution is to use a single camera, which enhances portability and makes it possible to carry out assessment in more confined spaces [24] such as the home or clinic. Use of a commercial RGB-D sensor system that integrates an RGB camera with a depth sensor in the same hardware is a promising approach.

Such an integrated system promises to help fill the current gap in infant movement assessment by providing a low-cost, compact platform that can be implemented repeatedly and longitudinally in the infant’s naturalistic environment, where movement repertoires are most likely to characterize the actual behaviors of the infant [19].

RGB-D sensors have been used in upper limb rehabilitation for adult patients post-stroke, as well as for analyzing balance recovery [25]. They have also been used in adult Parkinson’s disease patients to evaluate upper limb tasks [26], gait, and postural stability [27,28].

The current study aimed at recording infants’ upper body movements with a single, commercial RGB-D sensor; at tracking the 2D trajectories of selected points of interest (PoI) leveraging DeepLabCut [29], a well-established, open-source deep learning algorithm for pose estimation, i.e., generating 2D coordinates for tracked PoI; at obtaining the 3D PoI trajectories by applying a newly developed method; and finally at extracting GM metrics from the PoI 3D trajectories. This study proposes a novel method for assessing infants’ GM that features a simplified instrumental setup, suitable for home (or clinic) use.

This article presents in detail first the characteristics of the subjects involved and the experimental setup utilized, then how data were processed distinguishing what was already available from what was newly developed, and finally how GM metrics were obtained from 3D PoI trajectories. In presenting the results obtained, the clinical evaluation of two pediatric physicians with expertise in neurology and physiatry were taken into account.

## 2. Materials and Methods

The parents of eight infants recruited from the community volunteered to perform video recordings of their babies at their home. Infants, sitting in a baby seat covered with a green cloth to facilitate background exclusion during identification of infant PoI, were positioned on the floor in front of an RGB camera with an integrated depth sensor. The children’s natural movements were recorded for a target duration of three minutes at three different timepoints (3, 4, and 5 months from birth).

Infants used the same, washable seat throughout testing across different timepoints for consistency. Light conditions and interactions with humans were controlled to the degree possible in a home environment replicating the most natural conditions and guaranteeing the simplicity of the protocol.

Two expert physicians analyzed the recorded videos at each timepoint and were asked to report if they observed any cause for concern in the development of the infants.

The camera used for the recordings was a commercial RGB-D sensor (Intel RealSense D435, Intel, Santa Clara, CA, USA, combining a pre-calibrated RGB camera with 1280 × 720 native resolution and frame rate of ∼30 fps with a depth sensor generating depth-coded images with 640 × 480 native resolution and frame rate of ∼30 fps). Each pixel of the depth image had an intensity proportional to the distance of the surfaces in the image from the camera. Depth images were generated by the stereo vision of two infrared sensors mounted on the device with the left sensor used as point of view. If a region is seen only by the left sensor, the resulting depth image in that region remains black (“black area”). RGB and depth images were pre-calibrated; however, a residual misalignment between the two remained.

The markerless motion tracking software used in this study was DeepLabCut (Swiss Federal Institute of Technology, Lausanne, Switzerland) [29], an open-source toolkit for pose estimation in which a training set of images is manually labeled and returns the x, y coordinates of the tracked points along with a confidence level, varying from 0 (lowest confidence) to 1 (maximum confidence). DeepLabCut provides a framework for supervised, deep learning to tune an existing, high-performance convolutional neural network (ResNet, Microsoft Research, Redmond, WA, USA) to the features of a specialized dataset to produce a high level of recognition accuracy.

A sequence of steps was implemented to reconstruct the time series of the 3D coordinates of the selected points from the recorded RGB-D videos and to extract the associated GM metrics (Figure 1).

All blocks are explained in detail in the following sections.

### 2.1. RGB and Depth Images Acquisition and Time Alignment Refinement

The images acquired with the RGB camera and the depth sensor required refinement of the time alignment with respect to that obtained using the manufacturer’s proprietary software, since neither frame rate is exactly constant. The timestamps provided by the acquisition software were used for this purpose. Three alignment scenarios were, each of them requiring a different countermeasure:

The timestamp of an RGB image was closer to one or more RGB image timestamps than the closest depth image timestamp. Countermeasure: a gap of the proper number of frames was inserted in the sequence of depth frames;The timestamp of a depth image was closer to one or more depth image timestamps than the closest RGB image timestamp. Countermeasure: a gap of the proper number of frames is inserted in the sequence of RGB frames;The difference between the RGB and depth image timestamp was less than half the duration of the nominal sampling interval (<17 ms). The two frames were considered time aligned.

All gaps generated were then filled by applying a cubic spline interpolation.

### 2.2. 2D Tracking Algorithm

RGB images were converted into video files using ImageJ (National Institute of Health, Bethesda, MD, USA) [30] and fed to the DeepLabCut (Swiss Federal Institute of Technology, Lausanne, Switzerland) image processing tool. The tracking software was trained to identify six PoIs on the infant’s upper body: left and right shoulders (LS and RS), elbows (LE and RE), and wrists (LW and RW). All PoIs were manually labeled on 10% of the video frames (identified by DeepLabCut using a k-means algorithm that selected frames based on pixel characteristic variability) to create a training set.

After the network was trained and validated, DeepLabCut provided the PoI positions in all RGB frames, together with their confidence levels. When a PoI was occluded, its position was provided with a low confidence level. A recognition network was trained individually for each infant video to achieve the greatest possible accuracy prior to the association of RGB and depth coordinates of PoI.

### 2.3. Depth Reconstruction and 3D Coordinates Estimation

The 3D position of PoIs tracked in the RGB images was obtained by developing a method which exploits the depth sensor recordings. The location in the depth image corresponding to that of a tracked PoI in the concurrent RGB image was determined after addressing the three possible causes of incorrect or undefined PoI 3D positions:

the RGB location of a tracked PoI falling over the “black area” in the corresponding depth image, therefore lacking depth information (Figure 2a);PoI occlusions corrupting the estimation of PoI 3D positions. Since the tracking algorithm determines PoI locations exclusively from RGB information, the estimated location of a PoI could fall over a body segment covering the PoI in the RGB image (as when the head covers a shoulder). In such cases the estimate of the relevant depth coordinate would be affected by an error equal to the distance, along the depth direction, between the surfaces of the two body parts. To compensate for this error, the prediction confidence level values provided by the tracking software were used. The depth values obtained for frames with a confidence level lower than 0.6 were not considered (Figure 2b);a residual spatial misalignment between RGB and depth images causing errors in the estimation of the tracked PoI depth coordinate. Such a misalignment is responsible for errors in estimating depth coordinates when a tracked PoI is near a substantial depth discontinuity (Figure 2c). To compensate for the consequences of this error, the following procedure was implemented: the first derivative of the PoI depth coordinate was calculated; when its value was higher than a threshold value set based on the physical limits of the subject motion speed, the relevant depth value was removed.

All resulting depth coordinate gaps were then filled by applying a cubic spline interpolation.

**Figure 2 sensors-22-07426-f002:**
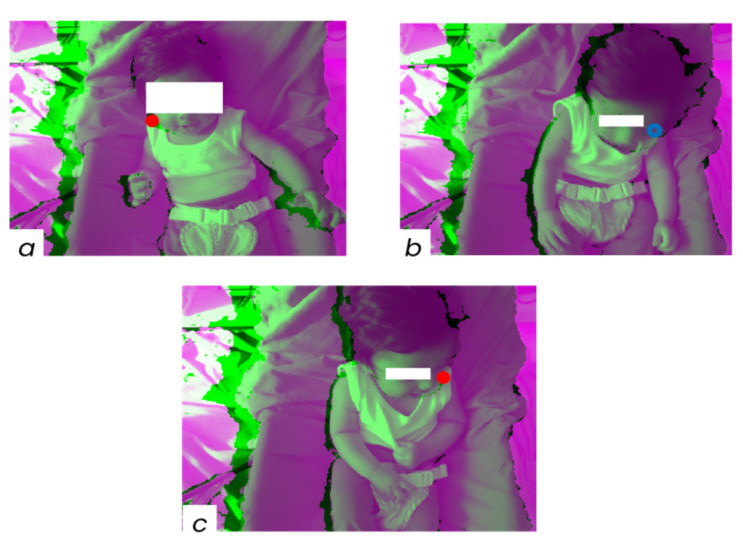
Issues causing undefined 3D PoI positions: (**a**) RS falling on the “black area”, (**b**) occlusion of the LS from the head, and (**c**) residual spatial misalignment between RGB and depth images. Subjects are made unidentifiable by using white patches. The coloured circles represent PoIs.

Finally, the identification of PoIs in the RGB images is conditioned by the way the PoI area is seen by the camera. Depending on the RGB frame, a single PoI may be identified in different areas of the infant’s body surface (Figure 3).

The performance of the markerless method described above was evaluated both on a physical model [31] as well as on real infants [32].

### 2.4. Kinematic Parameters and Metrics Estimation

From PoI 3D trajectories, the following metrics for quantifying GMs were estimated [16]:area in which the trajectories of the wrists differed from the moving average for the same trajectories, normalized with respect to the length of the moving average window (two seconds);area in which the trajectories of the wrists were outside of the standard deviation of the moving average for the same trajectories, normalized with respect to the samples in which the trajectories were outside the standard deviation (no information regarding the normalization was provided in the reference work);periodicity in the wrist trajectories;area in which the speed profiles of the wrists differed from the moving average for the velocity profiles, normalized with respect to the length of the moving average window (2 s);area in which the speed profiles of the wrists were outside of the standard deviation of the moving average for the velocity profiles, normalized with respect to the length of the moving average window (2 s);periodicity in the wrist velocities;the skewness of the velocities of the wrists;the cross-correlation of accelerations between left and right wrists.

In addition, we estimated the range of motion (ROM) of the elbow angle (EA), defined as the angle between the forearm segment and the upper arm segment.

To limit the influence of extended intervals of lack of upper limb movements to the estimated parameters described above, bouts of activity were introduced. The time intervals during which the infants’ wrists were moving were extracted from the rest of the acquisition. Bouts were defined as intervals of time characterized by wrist speed higher than a fixed threshold (5% of the wrist maximum velocity).

The blocks A, C, and D of Figure 1 were implemented in MATLAB R2021b (The MathWorks Inc., Natick, MA, USA).

## 3. Results

The two expert physicians involved in the study evaluated the RGB videos of the infants at 3, 4, and 5 months from birth to identify any features raising concern that an infant might not be typically developing (TD). Not all videos were commented on, but an overall evaluation of each infant was provided. The two physicians agreed that four infants (S1, S5, S7, and S8) appeared to be TD, and that one infant (S2) did show signs of possible atypical development. The physicians did not agree on the evaluation of the remaining three infants (S3, S4, and S6). Table 1 provides a complete description of the evaluations. The values of the nine GM metrics estimated from the upper body 3D PoI kinematics obtained by applying the proposed markerless method are reported in Figure 4 for each infant at each timepoint for left and right sides, both separately and together. To link the clinical evaluation to the metrics extracted, the range defined by the values found for those infants not suggesting atypical development was grayed in each plot.

Figure 5 and Figure 6 show the cross-correlation between left and right wrists accelerations and the range of motion of the elbow angle for each subject at each timepoint (3, 4, and 5 months), respectively.

Figure 7 shows the mean and standard deviation of the bout durations for each subject at each timepoint (3, 4, and 5 months) together with the number of bouts and movement duration, calculated as percentage of the acquisition time. Infants not suggesting atypical development (grayed) appeared to move their arms more than the other infants especially at the 4 and 5 month time points.

Numeric values are reported in Appendix A.

## 4. Discussion

The recording and the analysis of an infant’s upper body movements in a familiar environment has been a difficult and challenging task due to both technological and environmental factors. The technology used in previous studies has shown some limitations, since it is normally developed for and applied to the analysis of adults’ or children’s movements [33]. Markerless methods for the analysis of human motion have opened new possibilities for movement analysis, although initially it was mainly in two dimensions [20,21,34]. The analysis of infants’ movements may benefit substantially from markerless methods given the problems normally encountered in securely and safely positioning markers on their small body segments.

The low cost RGB-D cameras currently available in the consumer electronics market allow extending markerless techniques to 3D movement analysis without increasing the complexity of the experimental setup, a factor that allows leverage of the technique in environments outside the lab and promotes repeated measurement over time. The latter observation is of primary importance when infants’ movements are studied. Sensorimotor integration occurs rapidly in the first months of life through a process of activity-dependent neuronal modeling [35]. More frequent, routinized monitoring of infants’ movement in the convenient and familiar environment of the home increases the likelihood that infants who display abnormal movement repertoires will be identified promptly and interventions to prevent loss of neural connections and their specific functions instituted [7].

In this work we applied a markerless method to the RGB images recorded from a commercial RGB-D camera and used selected upper body PoIs extracted from the RGB video frames together with the recorded depth information to reconstruct 3D PoI kinematics [31] from which both some novel and already published metrics were calculated. The metrics used in this study were originally proposed to quantify GM [16,36], given the demonstrated power of GM assessment to predict the development of movement disorders very early in infancy [11]. Notably, since the key requirement of our approach to infant screening for neuromotor delay was that measurement be easily carried out in an informal environment such as the home, we did not attempt to replicate the General Movements Assessment in our protocol. For example, our infants were videoed in whatever attire their parents had chosen for the temperature in their homes, they were seated in a standard infant seat, versus lying supine, and videoed from the front using a commercial camera tripod versus from overhead requiring a special, suspended camera apparatus. The shift in infant posture likely caused the trends we calculated for GM parameters to vary from those reported by [36] for infants from three to five months of age.

Due to the small size of our sample, it was not possible to conduct meaningful statistical analyses. Rather, we describe visualized trends across three-, four-, and five-month measurement timepoints. Refer to Figure 4 for plots of parameters 1–7, to Figure 5 for a plot of parameter 8, and to Figure 6 for a plot of range of motion of the elbow angle. Table 2 summarizes the relationship between observed GM patterns and parameters, as well as the expected fluctuation in parameter values from three to five months of age in both TD and those later identified with CP. However, it should be noted that large differences between TD and non-TD infants would not be expected in our sample, as there was no documented injury that would have classified any of our non-TD infants as at-risk. In most of the metrics, variability in the data made it difficult to compare to predictions. However, in two metrics, trends were consistent with literature (described below).

The area where the wrist trajectory differs from the moving average of that same trajectory is suggested to quantify the diversity and variability of GM with respect to fluidity and congruence (metric #1). No significant change in the metric is expected from three to five months of age in TD infants, while those who are not TD are expected to exhibit smaller areas. Consistent with expectations, at four months, all infants about whom at least one of the clinicians evaluating videos expressed concern (S2, S3, S4, and S6) had smaller metric values relative to typical development (S1, S5, S7, and S8). The metric is expected to continually decrease in non-TD infants during the three-to-five month window. We did not, however, observe this trend as some infants have large swings in values across the timepoints.

The cross-correlation of acceleration between left and right wrists (metric #8) also showed trends consistent with previous work. This metric is associated with the observed characteristics of similarity and coordination of movement. TD infants may be expected to display movements that are similar, coordinated, and synchronous in the three-to-five-month window. Non-TD infants are expected to demonstrate the opposite movement pattern: dissimilar, uncoordinated, and asynchronous. This metric is expected to increase in TD infants between three and five months of age and to not increase during that time period in non-TD children. The metric values were higher at 5 months than 3 months in all infants for whom there was no concern (S1, S5, S7, and S8). However in S8, the cross-correlation at 4 months peaked sharply then regressed at five months but to a point still greater than the three-month cross-correlation. The cross-correlation measured for S6 (split concern) decreased monotonically between three and five months. Infants S2 (agreed concern) and S3 (split concern) virtually flatlined across all time points and logged cross-correlation metrics near the bottom of the cohort, well below the TD range at 5 months. Infant S4 (split concern) did not follow this pattern of decreasing or flatlined change over time. In summary, seven of the eight infants followed a pattern consistent with expectations.

We introduced elbow range of motion (Figure 6), a novel exploratory metric. Most infants’ ROM on the left arm fell within a narrow band changing little from three to five months. S1 (no concern) displayed very limited range at three months but increased to resemble the ROM of the cohort generally at four months. S2 (concern) demonstrated a ROM for the left arm among the highest in the cohort at three and five months but presented as the lowest at four months. The range of ROM angles was more diverse on the right side. S1 (no concern) was markedly low at three months and increased at four and five months, though not as much as had been noted on the left side. S2 (concern) lost range markedly at five months. A larger sample will be needed to determine how useful this metric will be for screening of non-TD infants.

Visualization of infants’ movement data suggest that the metrics are not independent. Environment influences that impact one impact others as well. Based on inspection of videos, several recommendations can be made for further work in this area. Better control of environmental factors might decrease variability in the data. While the protocol specifications were to have no one in the infant’s visual field during testing, it was difficult to enforce this in infants’ homes for all of the videos. In one case, a sibling approached the infant from the right causing him to lateralize in that direction. In another case, the infant had one hand in his mouth for a large portion of the video. For infants with this tendency, multiple capture sessions within the same day might be needed. For younger infants, the use of a baby seat was not optimal, as sitting posture was not fully developed. In the protocol, it was decided infants should use the same, washable seat throughout testing across different time points for consistency, sanitation, and to prevent infants from crawling away at older ages. Both the seat and infants’ upright position in the seat constrained their movements to an extent not experienced in the standard GM assessment protocol for which the quantification metrics we used were developed. Future studies should consider use of a standardized postural support method for younger seated infants.

In clinical practice, to determine whether an infant is typically developing or not, clinicians base their judgment on a full range of motor characteristics such as hand opening and closing and whether the infant brings his/her hands to the mouth. Characteristics such as these have assessment validity, but lie outside Prechtl et al.’s criteria for which the eight kinematic parameters proposed by [16,36] and colleagues (and applied in this study) were developed. Midline gaze, bringing hands to the midline, visual field preference, visual attention, social smile, and social engagement figure prominently among the criteria applied by our clinicians in applying their clinical discernment to our infant cohort. It would increase the power of 3D markerless movement assessment in infants to quantify observed clinical criteria such as those just enumerated to apply side-by-side with explicitly GM kinematics.

While evaluating videos, clinicians were also sensitive to the infants’ state. Characteristics of seeming to be sleepy, distraction from persons inevitably close to the home-based testing area, and, for infants who had not yet developed trunk control, being slumped to the side introduced ambiguity into the association of movement criteria with typical or pathological development. For example, infant S2, whom both clinicians flagged as exhibiting characteristics that caused concern, was documented as being slumped to the right at both the 3- and 4-month testing session. Being slumped to the side and concurrent lack of trunk control reasonably would predispose this infant to move asymmetrically. Notably, the cross-correlation between the left and right wrist accelerations of infant S2 were quite low, falling toward the bottom of the typical range at three months as defined by the range of values calculated for the cohort of infants whom clinicians evaluated as not of concern. Similarly, infant S6, for whom the highest left/right wrist acceleration cross-correlation was logged at three months, came in at the bottom of the not-of-concern cohort at four months but went on to log the lowest cross-correlation at five months of age. Clinician notes reveal that the infant had his finger in his mouth about 65% of the time at four months and for almost the entire duration of the video at five months. Clearly, when the infant’s spontaneous movements are restricted, as in the case of infant S2 who may not have been positioned so that both arms could move freely and as in the case of infant S6 whose side was (self-)constrained, the synchroneity, similarity, and coordination of movement on the left and right sides, summarized by the cross-correlation metric, is not representative of the infant’s actual movement characteristics. The constraint should be remediated and, ideally, the test repeated.

## 5. Conclusions

This work has shown the feasibility of estimating GM metrics with a single low-cost RGB-D sensor. The simplicity and portability of the proposed markerless protocol allows its use as a screening tool at home or any familiar environment and further makes it possible to avoid clinical environments which are artificial from a child’s perspective, and hence challenging for the assessment of true neurodevelopmental performance.

Compared with previous research, this article aimed to characterize GM without markers attached to the infants’ skin, which might interfere with infants’ spontaneous movements and consequentially affect their behavioral state. In addition, this markerless system provides 3D coordinates of each PoI, and is significantly advantageous over 2D motion capture when dealing with out of plane rotations and allowing more reliable characterization of GMs. Thanks to depth information provided by the RGB-D sensor, this protocol is able to deal with PoI occlusions that occur when using single-camera motion analysis. Our markerless system was designed especially for a home environment. This focus could be very beneficial for enhancing screening of neurodevelopmental disorders particularly for infants and families in rural and remote areas, a population with reduced health services. Due to the small size of our sample, it was not possible to conduct meaningful statistical analyses. For this reason, future studies will be devoted to validating the proposed protocol on a larger number of infants for testing its use in clinical practice.

## Figures and Tables

**Figure 1 sensors-22-07426-f001:**
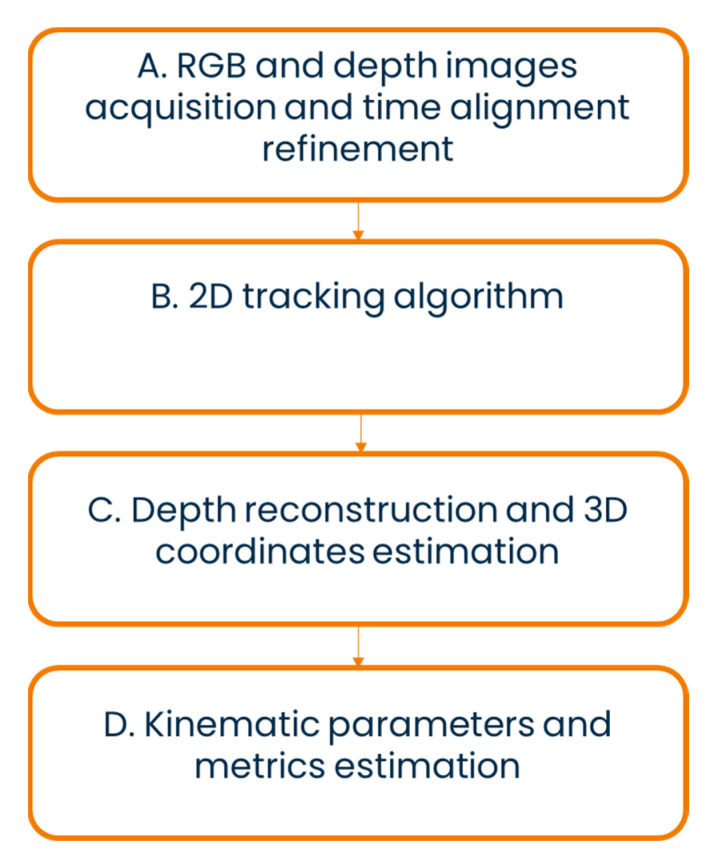
Block diagram of the proposed markerless-based method.

**Figure 3 sensors-22-07426-f003:**
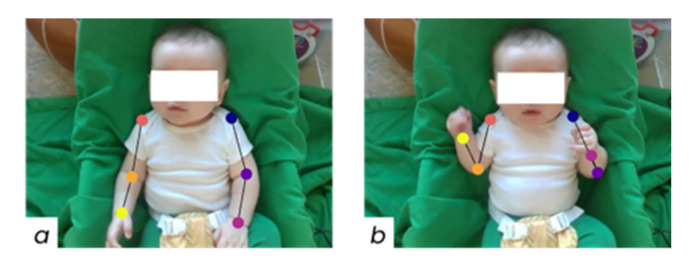
(**a**) RE is identified in the middle between epicondyles and (**b**) RE is positioned on the medial epicondyle. Subjects are made unidentifiable by using white patches. The coloured circles represent PoIs.

**Figure 4 sensors-22-07426-f004:**
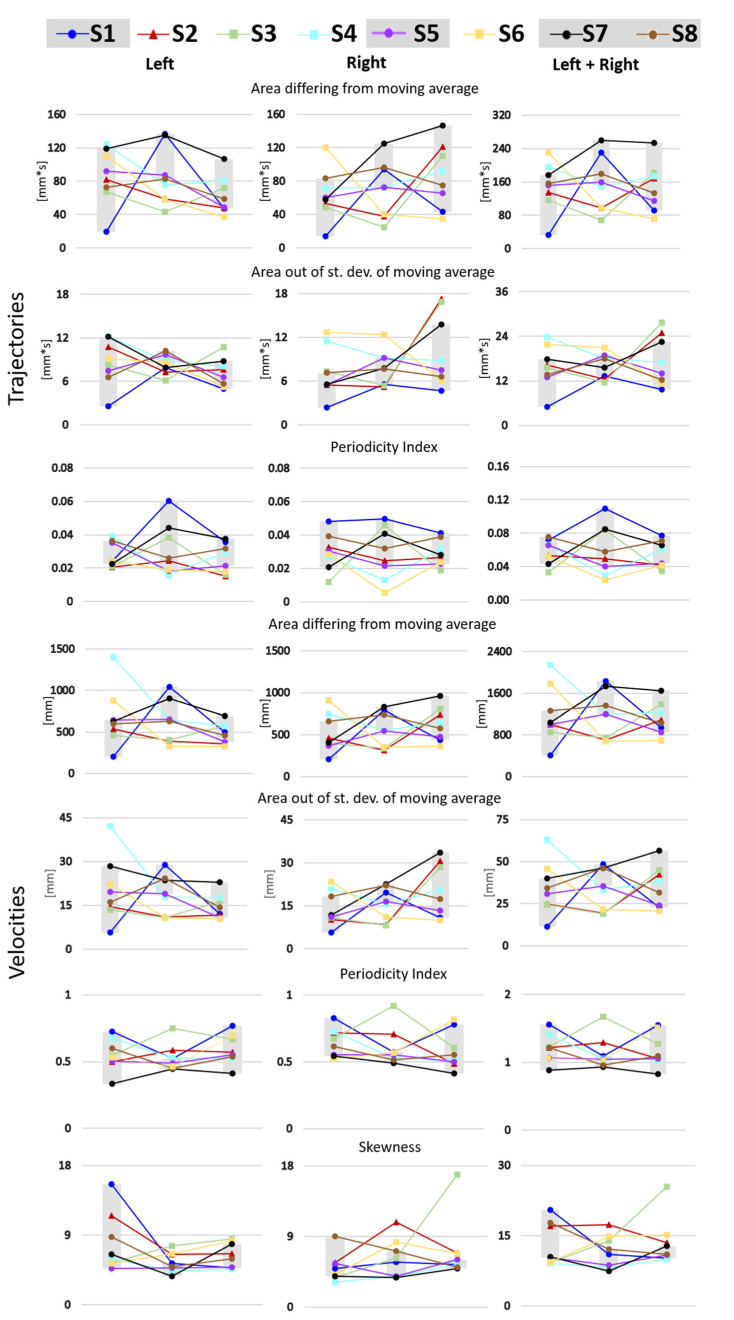
Metrics obtained from wrist trajectories and velocities for each infant at each timepoint (3, 4, and 5 months). Infants not suggesting atypical development to both physicians are identified with circles, infants raising the concern of both physicians are identified with triangles, and infants differently evaluated by the physicians are identified by squares. Infants not suggesting atypical development define the gray interval at each time point.

**Figure 5 sensors-22-07426-f005:**
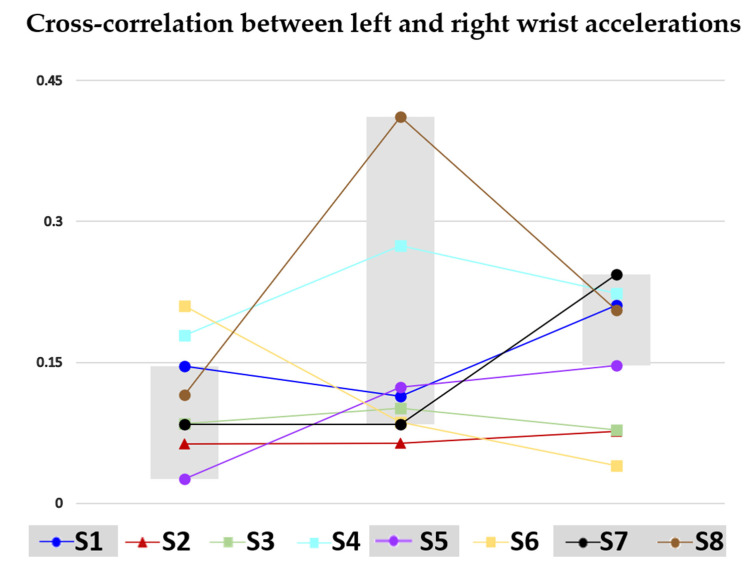
Cross-correlation between the left and right wrist acceleration for each infant at each timepoint (3, 4, and 5 months). Infants not suggesting atypical development to both physicians are identified with circles, infants raising the concern of both physicians are identified with triangles, and infants differently evaluated by the physicians are identified by squares. Infants not suggesting atypical development define the gray interval at each time point.

**Figure 6 sensors-22-07426-f006:**
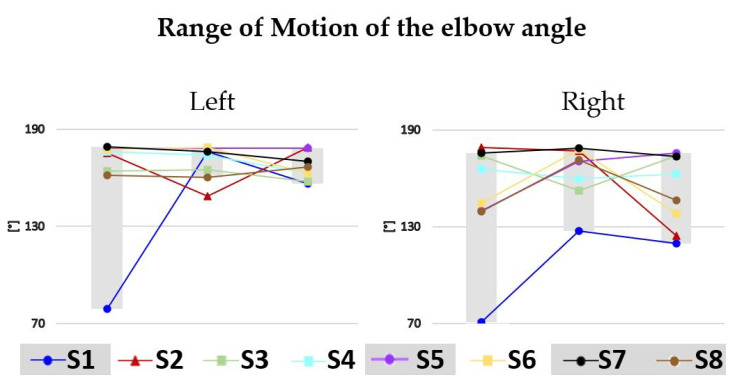
Range of motion of the elbow angle for the left and right side for each infant at each time point (3, 4, and 5 months). Infants not suggesting atypical development to both physicians are identified with circles, infants raising the concern of both physicians are identified with triangles, and infants differently evaluated by the physicians are identified by squares. Infants not suggesting atypical development define the gray interval at each time point.

**Figure 7 sensors-22-07426-f007:**
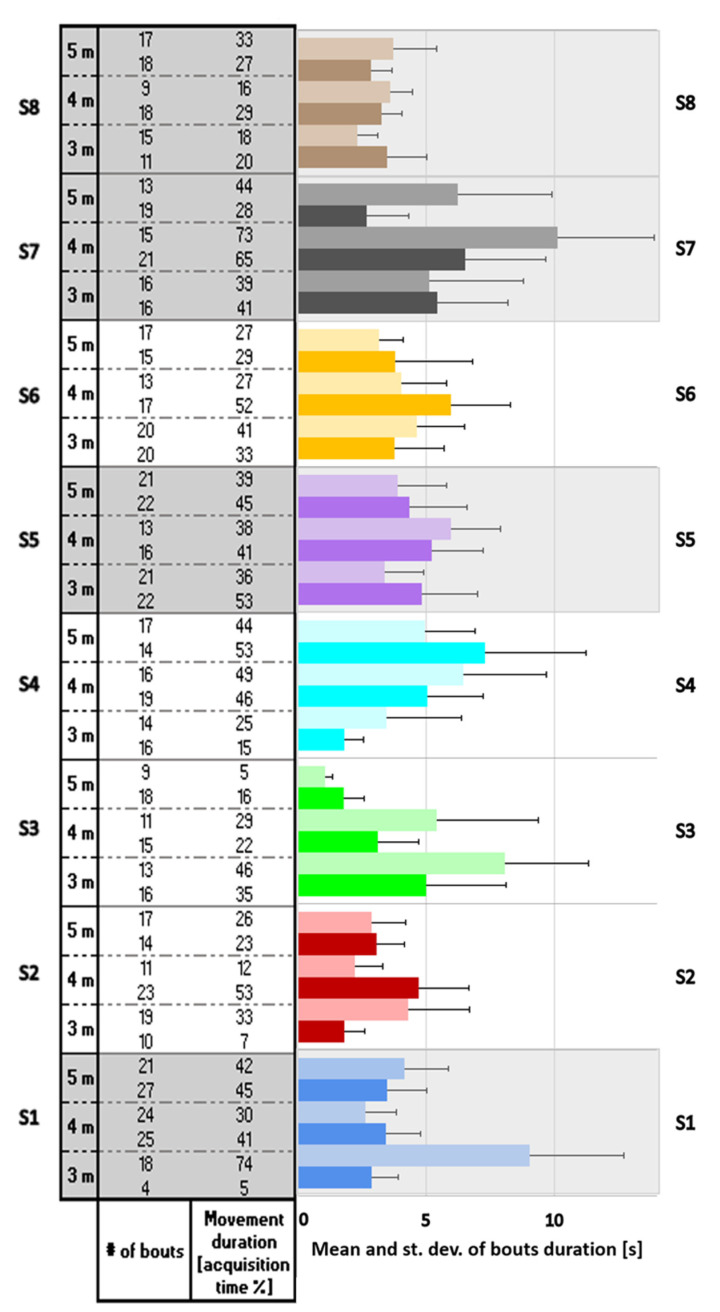
Mean and standard deviation of bouts duration for each infant at each timepoint (3, 4, and 5 months). The left side is the lighter color while the right side is the darker one. The number of bouts and the movement duration, calculated as percentage of the acquisition time, are reported in the table on the left. Subjects not suggesting atypical development are grayed.

**Table 1 sensors-22-07426-t001:** Report of the evaluation of the RGB videos performed by two expert physicians (A and B); subjects not suggesting atypical development are grayed.

*Sub#*	*Clinician*	*3 Months*	*4 Months*	*5 Months*	*Overall Evaluation* No: *Nothing Here Suggests the* *Infant Is Not Developing Typically* Yes: *I Did Observe Some Features that Raise Concern*
S1	A	-	-	-	No
	B	*lots of midline gaze*	*midline/R gaze but toddler on R*		No
S2	A	-	-	*Slow upper limb movements;* *no hands to mouth and midline;* *opens hands;* *thumbs frequently adducted*	Yes
	B				Yes
S3	A	-	-	-	No
	B				Yes
S4	A	-	-	-	No
	B	*decreased fidgety movements;* *subtle R hand preference?* *more fidgety movements on R*	*decreased fidgety movements;* *subtle R hand preference?* *more fidgety movements on R*	-	Yes *Lots of midline hand clasping and midline gaze preference at all ages*
S5	A	-	-	-	No
	B	-	-	-	No *Hands at midline;* *great gen and fidgety movements*
S6	A	-	-	-	No
	B	*non social smile; midline grasp*	*social smile; L fingers in mouth 65% of video;*	*L fingers in mouth entire video; no clear fidgety movements*	Yes
S7	A	-	-	-	No
	B	*Great visual attention; great gen and fidgety movements*	-	-	No *Grabbing toes; sucking on fingers; social*
S8	A	-	-	-	No
	B	-	-	-	No *Appears sleepy;* *improved visual attention and social engagement;* *good general movements;* *fingers or thumb in mouth*

**Table 2 sensors-22-07426-t002:** Mapping of observed patterns of general movements at 3–5 months of age to kinematic parameters. Adapted from ref. [16].

Metrics			Movements				
#	Class	Description	Aspect	Observed TD Characteristics	Measured TD Characteristics	Observed Non-TD Characteristics	Measured Non-TD Characteristics
1	Trajectories	Area where wrist trajectories differ from the moving average of the same trajectories	Diversity and Variability	Fluid and congruent	No significant change in area	Chaotic	Area smaller than TD, continues to diminish
2	Trajectories	Area where wrist trajectories of are outside the SD of the moving average of the same trajectories	Diversity and Variability	Multi-facetted	Smaller area, less diversity at 3 months (Increases after 5 months)	Poor-repertoire, spastic	Area smaller than TD, continues to diminish
3	Trajectories	Periodicity in the wrist trajectories	Unpredictability and Complexity	Fidgety	Periodicity decreases with age	Poor-repertoire	Periodicity greater than in TD
4	Velocities	Area where the wrist speed profiles differs from their moving average	Diversity and Variability	Fluid and congruent	Area does not change	Chaotic	Area decreases
5	Velocities	Area where the wrist speed profiles are outside the SD of the moving average the speed profiles	Diversity and Variability	Fidgety	Variation in velocity is constant	Cramped	Variation in velocity continuously decreases
6	Velocities	Periodicity in the wrist velocities	Equability of Velocity	Fidgety	Periodicity does not change	Cramped or chaotic	Periodicity does not change
7	Velocities	Skewness of the velocities of the wrists	Velocity Distribution of the Movement	Slow, small in amplitude	Skewness increases with age	Cramped, spastic	Skewness already increased by 3 months relative to TD
8	NA	The cross-correlation of accelerations between left and right wrists	Similarity and Coordination of Movement	Similar, coordinated, synchronous	Cross-correlation increases	Dissimilar, uncoordinated, asynchronous	Cross-correlation does not increase

## Data Availability

Matlab code used in this project can be provided upon request to the corresponding author.

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
