# Peer review of "Characterization of Infants’ General Movements Using a Commercial RGB-Depth Sensor and a Deep Neural Network Tracking Processing Tool: An Exploratory Study"

_sensors, 2022, doi:10.3390/s22197426_

Round 1

Reviewer 1 Report

The main concern of the reviewer is that the authors need to introduce the technical novelties more clearly. The method is a combination of existing techniques, then what are the novelty of this paper?A paper on multiple modal survey is relevant to this paper, which needs to be discussed Human action recognition from various data modalities: A review

Reviewer 2 Report

This article seems to have done a lot of work, and although the algorithm seems to be relatively simple, the research topic is very meaningful and make sense.

However, the problem is that the structure of the article needs to be revised, the second chapter is confusing, and the experimental part lacks quantitative indicators.

Reviewer 3 Report

The goal of the paper is to develop the characterization of infants’ general movements using a  commercial RGB-Depth sensor and a deep neural network tracking processing tool. Overall, the paper presents the in-depth results of the study, but it lacks to provide sufficient details about the deep neural network tracking processing tool.

Authors should provide a clear goal of their research in the abstract. Currently, the abstract describes the work, but no clear statement would point out the paper's main goal.

In the abstract, the authors state, “A custom algorithm was developed to estimate the 3D PoI using depth data.” However, the text of the paper does not elaborate sufficiently on the custom algorithm since the authors state that they use “a well-established deep learning algorithm for motion tracking”. So, it is not clear if the algorithm is custom or well-established.

A clear elaboration of the research gap in terms of previous research and the paper's contribution in that direction is lacking in the introduction. It should be added with the references of previous work most relevant to the research gap.

The authors should elaborate on the paper's content at the end of the introduction.

The authors mention Matlab at the end of the paper, but it is not mentioned in the text, so this should be elaborated on.

It seems that the abbreviation ML has been used for different terms, both for machine learning (e.g. “In this work, we applied an ML method to the RGB images”) and for the term markerless (ML). This should be clarified.

The content of Figure 1 (Figure 1. Block diagram of the proposed ML-based method.) is oversimplified; it should be more detailed. Besides, is the ML-based term in the title of Figure 1 referring to machine learning of markerless?

There are no sufficient details about the algorithm in sub-chapter B. 2D Tracking algorithm.

Prepare the text in Lines 162-182 more professionally since using bullets here makes it hard to read.

Line 192. “The performance of the ML method described above”. Is this referring to machine learning or the markerless method?

Lines 207-208 – Authors already propose a new measurement here but do not provide information about it. Currently, from the text, it is not clear that the new measurement is being proposed.

Figures 4, 5 and 6 are obsolete in the text, and it would be better to present the data from the figures in the table and move it to Appendix. It is hard to follow the data in figures 4, 5 and 6 since it is prepared for each infant separately. Besides, the average of the presented metrics according to infants suspicious of cerebral parsley and those not suspicious should be compared in such a table. Another possibility is to present the figures with the mean values of the presented metrics for those two groups of infants.

The current version of the text Comparison Against Prior Work Relating Parameters to Observed General Movement Patterns is hard to follow. It is too long and refers to infants regarding colours on figures, which is hard for the reader to follow. The suggestion is to present the data from figures 4, 5 and 6 in the table in the appendix. This part of the text should also be shortened; it currently looks like the part of the project report and not the scientific article.

Discussion regarding the theoretical contributions in deep learning for movement classification is lacking.

Practical implications are more oriented towards the well-known details, such as recording infants and their positioning at home (which should be shortened or put in the research context), while practical implications of the potential usage of the deep neural network are missing.

Please, form the conclusion in the following manner:

First paragraph - summary of research and conclusion - e.g. In this paper...

Second paragraph - comparison with previous research

Third paragraph - short description of practical implications

Fourth paragraph - summary of paper limitations and further implications

Reviewer 4 Report

The research conducted by Balta et al. is of value and remarks an interesting application of technology to general practice. The markerless assessment is deeply explained in the material and methods section, and the procedures seems to be adequate and reproducible.

A comparison of the method presented against a gold standard diagnosis would have been desirable. In this regard, one evaluator reported valuable extra information about his observations while another was merely dichotomous. The impressions of a third evaluator would have been useful given the great discordance among the observers. However, this lack of diagnostic consensus among the experts underlines the need to include objective methods for the detection of cerebral palsy and thus puts in value the main objective of the present study.

This reviewer appreciates the honesty with which the methodology and limitations are described. Some of the comments made in this review are only a repetition of the limitations acknowledged by the authors, in an attempt to emphasize the search for solutions for the current and/or future projects.

My main concern about the present work is the lack of clarity in the discussion and conclusion sections. The discussion should confront the results obtained for each parameter with expectations based on previous studies. In this sense, the expected results for each variable and month are not being clearly stated and discussed. On the contrary, it seems that the authors were probing the different variables evaluated in search of a useful pattern for the detection of pathology. This type of search would be very useful if a much larger sample size and a reliable diagnosis by a gold standard method were available. In fact, a proposal for the future would be to establish the normative values in healthy infants for each month of their development and from them to find the variables that are most affected in patients with cerebral palsy. However, differences between biological and chronological age, as well as fluctuations in the developmental speed of infants could lead to errors in the prediction of the algorithm created.

The conclusions section does not summarize the findings of the research nor really show the practical application of the results, but only the future lines that the present investigtion opens. In this sense, it is expected that the authors work more on highlighting the value of their work and show more specific conclusions.

Another comments:

I suggest to reduce the number of abbreviations and to explain them in its first appearance (e.g., TD is only defined in table 2 and not in its first appearance in the main text, line 262).

Line 366: add a colon or remove "three"

As the authors acknowledge (lines 511-516), the selection of the sitting position for filming is problematic as it is not well developed at 3 months of age in TD.

Round 2

Reviewer 1 Report

Overall the method and the research problem is interesting.. However i have the following concerns:

1- RGBD camera is often not very accurate and often includes lots of holes in the depth map. How to handle this issue?

2-RGBD camera often cannot work well in outdoor scenes. Will this limit the application of the framework.

3-Authors may briefly discuss the multi-camera multiview setup, and describe the advantages of this method /direction over it.

4-Since this is a exploratory  study, it is good to discussion the weaknesses/failure cases of the framework.

5- The following RGBD human activity analysis works are relevant, which need to be discussed: UAV-Human: A Large Benchmark for Human Behavior Understanding with Unmanned Aerial Vehicles

Reviewer 2 Report

The fourth sentence of the abstract has a punctuation error.

The second chapter only has the result picture, lacks the detailed algorithm flow graph,  which leads to poor readability.

Reviewer 3 Report

Dear authors, I see that you have accepted the comments, and my suggestion is that the paper is now ready for the publication. 

Author Response

Thank you for your comment.